# Impact of the COVID-19 pandemic on intimate partner violence during pregnancy: evidence from a multimethods study of recently pregnant women in Ethiopia

Shannon N. Wood ![ORCID],[1] Robel Yirgu,[2] Abigiya Wondimagegnehu,[2] Jiage Qian,[1] Rachel Mait Milkovich ![ORCID],[1] Michele R Decker,[1,3,4] Nancy Glass,[4,5] Fatuma Seid,[6] Lensa Zekarias,[6] Linnea A. Zimmerman ![ORCID][1]

For numbered affiliations see end of article.

**Correspondence to**
Dr Shannon N. Wood;
swood@jhu.edu

## ABSTRACT

**Objectives** This multimethods study aimed to: (1) compare the prevalence of intimate partner violence (IPV) during pregnancy pre-COVID-19 and during the COVID-19 pandemic using quantitative data and (2) contextualise pregnant women's IPV experiences during the COVID-19 pandemic through supplemental interviews.

**Design** Quantitative analyses use data from Performance Monitoring for Action-Ethiopia, a cohort of 2868 pregnant women that collects data at pregnancy, 6 weeks, 6 months and 1-year postpartum. Following 6-week postpartum survey, in-depth semistructured interviews contextualised experiences of IPV during pregnancy with a subset of participants (n=24).

**Participants** All pregnant women residing within six regions of Ethiopia, covering 91% of the population, were eligible for the cohort study (n=2868 completed baseline survey). Quantitative analyses were restricted to the 2388 women with complete 6-week survey data (retention=82.7%). A purposive sampling frame was used to select qualitative participants on baseline survey data, with inclusion criteria specifying completion of quantitative 6-week interview after the onset of the COVID-19 pandemic, and indication of IPV experience.

**Interventions** A State of Emergency in Ethiopia was declared in response to the COVID-19 pandemic approximately halfway through 6-week postpartum interview, enabling a natural experiment (n=1405 pre-COVID-19; n=983 during-COVID-19).

**Primary outcome measures** IPV during pregnancy was assessed via the 10-item Revised Conflict and Tactics Scale.

**Results** 1-in-10 women experienced any IPV during pregnancy prior to COVID-19 (10.5%), and prevalence of IPV during pregnancy increased to 15.1% during the COVID-19 pandemic (aOR=1.51; p=0.02). Stratified by residence, odds of IPV during the pandemic increased for urban women only (aOR=2.09; p=0.03), however, IPV prevalence was higher in rural regions at both time points. Qualitative data reveal COVID-19-related stressors, namely loss of household income and increased time spent within the household, exacerbated IPV.

### Strengths and limitations of this study

► Natural experiment based on split of pre-existing cohort to examine differences in intimate partner violence (IPV) during pregnancy pre-COVID-19/post-COVID-19 restrictions.
► Large quantitative sample inclusive of all pregnant women residing within six regions of Ethiopia, covering 91% of the population.
► In-depth interviews to contextualise women's experiences with IPV during pregnancy during the COVID-19 pandemic.
► Groupings may not exactly estimate pre/post exposure and no women who were truly only exposed to IPV during the COVID-19 pandemic.
► Qualitative interviews only occurred within two regions and are not transferable to all pregnant women experiencing IPV during pregnancy.

**Conclusions** These multimethods results highlight the prevalent, severe violence that pregnant Ethiopian women experience, with pandemic-related increases concentrated in urban areas. Integration of IPV response and safety planning across the continuum of care can mitigate impact.

## INTRODUCTION

Pandemics exacerbate gender inequities, including intimate partner violence (IPV).[1–3] Evidence from previous epidemics and regional crises indicates heightened IPV via increased economic insecurity, social isolation and exposure to perpetrators, and more limited options for garnering support.[1] COVID-19 is no exception—12 out of 15 early studies indicate increases in IPV globally.[4]

The pregnancy and postpartum period are pivotal time points for assessing and responding to IPV, given health impacts to

mother and baby, and multiple points of contact with healthcare providers. Global estimates indicate that 2%–14% of women experience violence during pregnancy.[5 6] Pre-existing IPV may be sustained or exacerbated, and abusive partners may limit access to antenatal and postnatal care.[6] IPV during pregnancy incurs profound effects, including miscarriage, premature labour, low birth weight and maternal depression.[7–10] Timely response is thus critical for this vulnerable subpopulation.[5 10] Linkages to woman-centred care may decrease subsequent abuse.[10]

Violence prevention and response is critical within the Ethiopian context. Ethiopia Demographic and Health Survey (DHS) data indicate that 27% of ever-married women age 15–49 have experienced IPV within the past year and 4% experienced IPV during pregnancy.[11] In April 2020, Ethiopia declared a State of Emergency in response to the COVID-19 pandemic, with resultant lockdown measures including, but not limited to, physical distancing, school closures, prohibition of in-house visitation outside of family members, capacity limits and masking requirements on public transportation, and discouragement of within-country travel[12]; these measures may disproportionately impact urban residents due to population density, urban planning, and health-seeking behaviours.[13] Further, though essential to limit infection, such lockdown measures may exacerbate pregnant women's IPV experiences.[2 3] Early evidence from Amhara, Ethiopia indicated similar levels of IPV pre-COVID-19 and during COVID-19,[14] while 18% of antenatal care attendees in Addis Ababa reported perceived increases in violence during pregnancy since the onset of the pandemic.[15] Both studies, however, were limited in generalisability due to cross-sectional design and sampling considerations.

While global evidence indicates increases in IPV, to date, no studies aimed at understanding the impact of the COVID-19 pandemic or related response measures have examined violence during pregnancy within the Ethiopian context.[4] Ethical standards advocate for continued IPV monitoring using existing study infrastructures if the research aims to understand the magnitude of IPV burden and linkages with social/economic factors; informs response efforts; and meets ethical obligations.[1 16] Using an existing national cohort of recently pregnant women and rigorous ethical standards, this study aimed to examine how the COVID-19 pandemic affected pregnant women's experiences with IPV through (1) comparison of IPV prevalence pre-COVID-19 and post-COVID-19 using quantitative data and (2) contextualisation of women's experiences of IPV during the COVID-19 pandemic.

## METHODS
### Overall study design
This multimethods analysis is situated within the Performance Monitoring for Action (PMA)-Ethiopia cohort study, a collaboration between Johns Hopkins Bloomberg School of Public Health (JHSPH), Addis Ababa University (AAU) and the Ethiopian Federal Ministry of Health (FMoH). PMA-Ethiopia collects quantitative data on a cohort of 2879 pregnant women at pregnancy (any gestation), 6 weeks, 6 months and 1-year post partum. Enrolment into the cohort began in October 2019. The full protocol for PMA Ethiopia is detailed elsewhere.[17] The present analysis uses quantitative 6-week postpartum data to allow for most comprehensive measurement of violence throughout the entire pregnancy period. Following 6-week interview, in-depth interviews contextualised experiences of IPV during pregnancy, among a subset of participants (n=24); the qualitative phase was not a component of the original quantitative study design, and the quantitative data did not inform qualitative analyses and vice versa.[18]

### Quantitative participants
All pregnant women residing within six regions of Ethiopia that covers 91% of the population were eligible for the cohort study (n=2868 completed baseline survey). Quantitative analyses were restricted to the 2388 women with complete 6-week survey data (retention=82.7%). A State of Emergency in Ethiopia was declared in response to the COVID-19 pandemic approximately halfway through fielding the 6-week postpartum interview (8 April 2020), thus enabling a natural experiment. Specifically, 1405, 6-week interviews were conducted before the onset of the COVID-19 pandemic; the remaining 983 occurred after COVID-19 emergency lockdown procedures eased in early June and data collection was able to resume.

### Quantitative measures
IPV, the primary outcome of interest, was measured via the 10-item Revised Conflict and Tactics Scale,[19] which asks about specific violence behaviours at any time during pregnancy, per best practices for violence research.[18] Three dichotomous violence measures were examined: (1) any IPV, (2) any physical IPV, (3) any sexual IPV; affirmative response to any behaviour was classified as IPV experience. Physical and sexual IPV were derived from the following items: (1) physical IPV: 'push you, shake you or throw something at you;' 'slap you;' 'twist your arm or pull your hair;' 'punch you with his fist or with something that could hurt you;' 'kick you, drag you or beat you up;' 'try to choke you or burn you on purpose;' 'threaten or attack you with a knife, gun or other weapon' and (2) sexual IPV: 'physically force you to have sexual intercourse with him when you did not want to;' 'physically force you to perform any other sexual acts you did not want to;' 'used threats or pressure to make you have sex when you did not want to.' Any physical and any sexual IPV were not mutually exclusive.

The primary exposure variable of interest measures pre/post exposure to COVID-19 restrictions, a binary variable, indicating whether a woman completed her 6-week postpartum survey before (prelockdown) or after

(postlockdown) 8 April, when the State of Emergency was declared and data collection paused.

Analyses were stratified by urban/rural residence given the differences in COVID-19 lockdown measures. Sociodemographic variables explored as adjustment variables were chosen on a conceptual basis and included region, household wealth, age, parity and education; all were examined in categorical form.

## Quantitative analysis

Descriptive statistics examined the distribution of sociodemographic characteristics by pre/post exposure; design-based F statistics assessed whether distributions were similar. Venn diagrams classified violence experiences during each time point. Next, bivariate distributions of each type of violence were examined by exposure, overall and stratified by urban/rural residence. Bivariate (not presented) and multivariable logistic regression models were used to examine differences in violence experience pre/during the COVID-19 pandemic and residence; for adjusted models, correlations between covariates were examined for multi-collinearity, with only residence, age and education retained within final models. All analyses were conducted in STATA V.16, with statistical significance set at $p < 0.05$, and accounting for complex-survey design.

## Qualitative participants

To contextualise the impact of the COVID-19 pandemic on women's experiences of violence during pregnancy, an explanatory qualitative phase was conducted following principles for qualitative descriptive design.[20] Specifically, a purposive sample of 24 women identified in the survey data to have experienced IPV were invited to participate semistructured qualitative interviews in the month following 6-week quantitative data collection in Oromiya and Southern Nations, Nationalities, and People's (SNNP) regions; regions were selected based on high IPV during pregnancy from baseline survey and feasibility. At the 6-week survey, participants were consented for potential follow-up specific to partner-related items. A purposive sampling frame was used to select participants on baseline survey data, with inclusion criteria specifying: (1) completion of quantitative 6-week post-COVID-19 interview; (2) indication of IPV experience via quantitative data. These eligibility criteria generated a sampling frame of 17 IPV survivors in Oromiya and 18 IPV survivors in SNNP. From this purposive sampling frame, participants were randomly selected and data collection continued until feasible sample size was met (n=14 Oromiya; n=10 SNNP).[21]

## Qualitative data collection

Training for the qualitative phase preceded data collection with focus on probing, ethical principles for IPV research, and research team protections. Semistructured interview guides focused on women's experiences with IPV and IPV services. Participants were called prior to interview for scheduling considerations. All interviews lasted approximately 25–30 min. Four trained interviewers used a structured note-taking tool to allow probing of experiences, while permitting rapid analysis for timely results.[22] Interviews were conducted in Amharic, Afan Oromo or local languages of SNNP with the help of local translators. Immediately postinterview, interviewers typed and translated field notes.

## Qualitative analysis

Two researchers trained in qualitative analysis coded 24 structured notes using Atlas.ti software. Inductive thematic analysis was used to identify themes and subthemes and to create an initial set of codes. Dual coding and retrocoding were used to enhance agreement between coders. Coders also met and collaborated regularly with the field research team to discuss and clarify interpretation. Coding was complete when saturation of themes was achieved[23]; illustrative quotes were then downloaded from Atlas.ti and organised in matrices of code themes that were organised by IPV experience.

## Patient and public involvement

Community members are not directly involved in PMA-Ethiopia research, however, a project advisory board, including members of the FMoH, health providers and key stakeholders, are consulted about country-specific priorities, including IPV, during survey development. Results are disseminated to the advisory board to inform action.

## RESULTS
### Quantitative results

Demographic characteristics of study participants by pre/post exposure are presented in table 1. Significant differences were observed for parity only, where women of higher parity were more likely to be interviewed pre-COVID-19.

Figure 1 displays overlap of types of violence experienced by women during pregnancy, by exposure. At both time points, sexual IPV ($7.4\%_{\text{pre-COVID-19}}$; $9.8\%_{\text{during-COVID-19}}$) was higher than physical IPV ($5.0\%_{\text{pre-COVID-19}}$; $7.8\%_{\text{during-COVID-19}}$) and often occurred in isolation.

Bivariate and logistic regression results comparing pre-COVID-19 IPV to during COVID-19 IPV, overall and by residence, are presented in table 2. One in 10 women experienced any IPV during pregnancy prior to COVID-19 (10.5%), and this proportion increased to 15.1% during the COVID-19 pandemic (adjusted odds ratio (aOR) during COVID-19 compared with pre-COVID-19=1.51; 95% CI=1.06 to 2.15; p=0.02). When stratified by residence, odds of IPV during COVID-19, compared with pre-COVID-19, increased for urban women (aOR=2.09; 95%CI=1.10 to 3.96; p=0.03), while odds slightly attenuated for rural women, but the difference did not achieve statistical significance (aOR=1.43; 95% CI=0.96 to 2.13; p=0.08). IPV prevalence was consistently higher among

**Table 1** Characteristics of 6-week interview participants by exposure (n=2388)

| Demographic characteristics | Pre-COVID-19 6-week interview (n=1405) | During-COVID-19 6-week interview (n=983) | P value |
|---|---|---|---|
| | n (row %) | | |
| Region | | | 0.28 |
| Tigray | 107 (61.5) | 66 (38.5) | |
| Afar | 23 (50.0) | 23 (50.0) | |
| Amhara | 305 (61.4) | 191 (38.6) | |
| Oromiya | 573 (55.6) | 459 (44.4) | |
| SNNP | 332 (59.6) | 225 (40.4) | |
| Addis Ababa | 45 (54.4) | 38 (45.6) | |
| Residence | | | 0.14 |
| Urban | 320 (61.0) | 205 (39.0) | |
| Rural | 1064 (57.2) | 797 (42.8) | |
| Household wealth | | | 0.23 |
| Lower | 531 (56.2) | 413 (43.8) | |
| Higher | 854 (58.2) | 589 (40.8) | |
| Age | | | 0.86 |
| 15–19 | 147 (56.5) | 114 (43.6) | |
| 20–29 | 736 (57.9) | 534 (42.1) | |
| 30–49 | 502 (58.6) | 355 (41.4) | |
| Parity | | | 0.001 |
| Nulliparous | 243 (50.3) | 238 (49.5) | |
| 1–2 | 538 (62.3) | 326 (37.7) | |
| 3+ | 604 (57.9) | 438 (42.1) | |
| Education | | | 0.95 |
| Never attended | 579 (57.8) | 422 (42.2) | |
| Primary | 554 (58.5) | 393 (41.5) | |
| Secondary or higher | 252 (57.4) | 187 (42.6) | |

SNNP, Southern Nations, Nationalities, and People's Region.

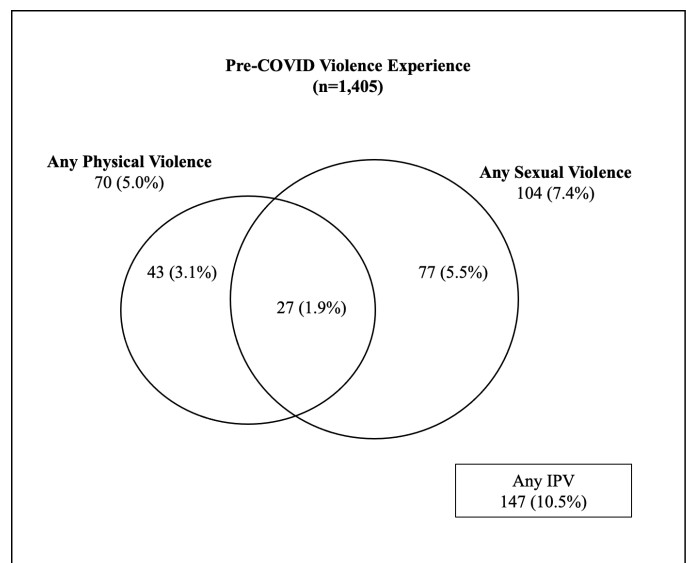
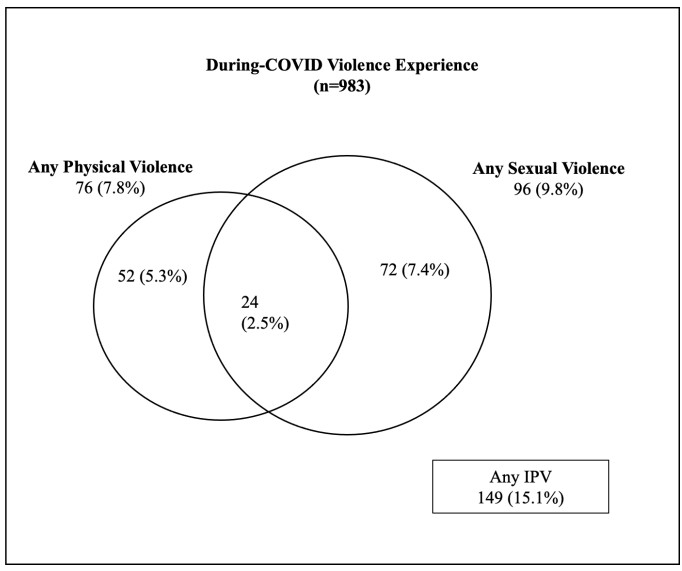

**Figure 1** Venn diagram of types of violence experienced during pregnancy, by pre-exposure/post-exposure.

**Table 2** Bivariate and logistic regression examining type of violence post-COVID-19, compared with pre-COVID-19, overall and by residence

| Type of violence | Overall (n=2388) | | | Urban (n=884) | | | Rural (n=1504) | | |
| --- | --- | --- | --- | --- | --- | --- | --- | --- | --- |
| | Pre (n=1405) n (%) | During (n=983) n (%) | aOR† (95% CI) | Pre (n=540) n (%) | During (n=344) n (%) | aOR‡ (95% CI) | Pre (n=865) n (%) | During (n=639) n (%) | aOR‡ (95% CI) |
| Any IPV | 147 (10.5) | 149 (15.1) | 1.51* (1.06, 2.15) | 30 (5.6) | 37 (10.9) | 2.09* (1.10, 3.96) | 103 (11.9) | 103 (16.2) | 1.43± (0.96, 2.13) |
| Physical Violence | 70 (5.0) | 76 (7.8) | 1.60* (1.07, 2.40) | 19 (3.5) | 27 (7.9) | 2.46* (1.18, 5.10) | 47 (5.4) | 49 (7.7) | 1.46 (0.90, 2.36) |
| Sexual Violence | 104 (7.4) | 96 (9.8) | 1.33 (0.88, 2.02) | 15 (2.8) | 16 (4.7) | 1.70 (0.59, 4.91) | 76 (8.8) | 71 (11.1) | 1.30 (0.83, 2.03) |

±p<0.10; *p<0.05; **p<0.01; ***p<0.001.
†aOR adjusted for residence, age, and education; parity omitted for multicollinearity.
‡aOR adjusted for age and education; parity omitted for multicollinearity.
aOR, adjusted odds ratio; IPV, intimate partner violence.

rural women at both time points (16.2% during COVID-19, 11.9% pre-COVID-19 for rural women vs 10.9% during COVID-19, 5.6% pre-COVID-19 for urban women).

In examining violence subtypes, physical IPV increased 60% during COVID-19, compared with pre-COVID-19, in the overall sample (aOR=1.60; 95% CI=1.07 to 2.40; p=0.02); this increase was greater than twofold in urban areas (aOR=2.46; 95% CI=1.18 to 5.10; p=0.02). Statistically significant changes by pre-exposure/post-exposure were not observed for physical violence in rural areas or for sexual violence.

## Qualitative results

Qualitative data indicate that the COVID-19 pandemic and implementation of local lockdown measures affected IPV within relationships differentially based on husband's job status. Specifically, women whose husbands experienced loss of income or employment due to the pandemic often reported that the COVID-19 pandemic put stress on their relationship and increased instances of physical violence. These stressors were described by an 18-year-old woman in Oromiya who had just given birth to her first child:

> Since he was not working, he didn't have money; it was difficult for us to pay the house rent, it was difficult to buy food. These problems created conflict, then he insulted and hit me.

Further, many women experienced an increase in verbal abuse due to COVID-19-related stressors. Several participants shared that the main reason for conflict prior to the pandemic was disagreement regarding household responsibilities, suggesting that the COVID-19 pandemic exacerbated this tension. This increase in household tension due to restrictions on movement was discussed by a 36-year-old woman with four children in rural SNNP:

> The violence increased because we stayed together at our house because there was movement restriction in our village so he couldn't go to the market to sell

goats and sheep. So, our income decreased a lot and because of that we had arguments most of the time.

The pandemic also changed the amount of time that spouses spent at home together, most often due to a husband's loss of employment. One 29-year-old participant from rural Oromiya disclosed that this gave her husband an opportunity to further control her activities and to engage in sexual violence at a high frequency:

> During corona time, he stays at home so he is happy with that because he can control all my day-to-day activities. No physical and psychological violence happened but sometimes there is sexual violence. It might be four times a week.

It should be noted that several participants stated that the pandemic had little or no impact on their relationships with their husbands, however, previous violence persisted throughout. An example of this endemic violence is described by a 37-year-old IPV survivor from rural Oromiya:

> We had a disagreement during this pregnancy. It was during coronavirus. But the cause was not related with corona. We disagreed due to children. The corona pandemic has nothing to do with our relationship.

In many instances, the violence described by women during pregnancy was severe— examples of violence severity, particularly in relation to physical violence, are described below.

> When I became pregnant, the part of my body he hits was changed but not the frequency. He used to beat every part of my body but when I became pregnant, he especially didn't hit me around the belly. Because he cares for the baby. He slaps my face with my own hand.—36-year-old IPV survivor with four children from rural SNNP

He doesn't care if I am pregnant or not. When he is very angry, he gets very emotional and throws at me whatever he finds, whether it is a chair or anything. I had stillbirth, the fetus died in the belly, though I had a spontaneous vaginal delivery. When he heard the baby died in my uterus, he got angry and thought I was the one who killed the baby. –18-year-old nulliparous IPV survivor from rural SNNP

When I was nine months pregnant; he hit me and threw me on the stone, then I went to my father's home. Immediately, my father's neighbors took me to hospital. I was admitted there for 15 days.— 18-year-old primiparous IPV survivor from urban Oromiya

## DISCUSSION

These multimethods results highlight the prevalent, severe violence that women experience at a critical, and often vulnerable, period during their reproductive lives. Using a natural experiment embedded in cohort of recently pregnant women in the six largest regions of Ethiopia, we found heightened IPV during the COVID-19 pandemic, compared with the pre-COVID-19 period, with greatest between-time impact concentrated in urban areas. Discrepancies by residence corroborate heightened cases and more COVID-19 preventive measures within urban areas.[12] Qualitative data reveal the mechanisms through which pandemic-related stressors operate to exacerbate IPV, namely loss of household income and increased time spent within the household.

More than 1-in-10 women experienced violence during pregnancy within Ethiopia ($10.5\%_{pre-COVID-19}$)—IPV during pregnancy reported by recently pregnant women in this study was substantially higher than 2016 DHS lifetime estimates (3.7%).[11] Consistent with the WHO Multi-Country Study on Women's Health and Domestic Violence and DHS,[6 11] women reported higher levels of sexual violence than physical violence—notably, Ethiopia is one of the few contexts globally where this pattern is observed. Continuous monitoring within national surveillance systems and large-scale cohort studies can assist identifying who is at risk for IPV during most recent pregnancy. Using best practices for violence-related research,[16 18] for example, timely linkage to woman-centred referrals, can bolster safety and minimise adverse outcomes.

Given distinct patterns of IPV by urban and rural residence, interventions must be tailored by locality and women's circumstances. The impact of the COVID-19 pandemic on violence during pregnancy was concentrated primarily within urban settings. Immediate intervention and connection to IPV referral services should focus on urban settings, prone to closures and economic impact. As time and household finances were indicated as stressors, temporary separation and seeking support from family members and neighbours may be a feasible strategy to minimise impact of violence—this safety strategy has been useful for women experiencing IPV in urban informal settlements of Nairobi and other low- and middle-income contexts where leaving the relationship is not feasible or socially acceptable.[24 25] Notably, however, IPV during pregnancy was highest for rural women at both time points ($16.2\%_{rural}$ vs $10.9\%_{urban}$ during COVID-19). Heightened prevalence within rural settings likely speaks to cohesive, patriarchal community norms promoting violence against women[26]; in order to ultimately reduce IPV, large-scale transformative community norms interventions, such as Communities Care or SASA!,[27 28] will be required.

This study is not without limitations—namely, these groupings may not exactly estimate pre/post exposure and we have no women who were truly only exposed to IPV during the COVID-19 pandemic. Further, our sample size was limited for additional subgroup analysis to examine women most at risk for IPV during pregnancy by pre-COVID-19 and during-COVID-19 time points. To maximise women's confidentiality and bolster safety in line with best practices for violence-related research,[18] interviewers were instructed to not conduct either quantitative or qualitative interviews within the presence of a partner; given COVID-19 lockdown measures and potential for controlling behaviours within abusive partnerships, our results may be an underestimate. Lastly, qualitative data collection occurred only within two of the six regions based on baseline prevalence of IPV during pregnancy—this purposive sampling strategy may have excluded women needing IPV support during the COVID-19 pandemic and is not transferable to all pregnant women experiencing IPV in Ethiopia.

While IPV during pregnancy increased overall during the COVID-19 pandemic, violence and related gender-based power disparities were prominent prior to the onset of the pandemic—accordingly, IPV prevention and response efforts in Ethiopia must be sustained in the postpandemic era. We offer two concrete recommendations for maternal health providers. First, while antenatal care is a critical intervention point for identifying and providing care for women experiencing IPV during pregnancy, providers across the maternal and neonatal continuum of care must similarly provide support. Integration of IPV screening and psychological care into postnatal care is critical given links to postpartum depression.[29 30] Clinic-based aids, including WHO recommendations for clinical IPV identification and linkage to health and economic referrals, can be valuable in training providers to support survivors.[31] Second, the use of safety decision-aids by providers across the continuum of care could assist in helping women assess their circumstances and level of danger, and create safety plans tailored to their situations. Safety planning with trained community health workers was found to increase safety preparedness in Kenya,[32] and may be similarly valuable in rural areas of Ethiopia with trained Health Extension Workers. Clinic-based interventions to reduce and address IPV should not occur separately from other health services, but as part of larger community-led behavioural change programmes

with influential community members on targeting harmful gender norms and IPV, in order to ultimately empower women and girls. The COVID-19 pandemic has alerted the global community to pervasiveness of violence against women—continued momentum and investment is needed to mitigate its harmful impact.

**Author affiliations**
[1]Population, Family and Reproductive Health, Johns Hopkins Bloomberg School of Public Health, Baltimore, Maryland, USA
[2]Addis Ababa University School of Public Health, Addis Ababa, Ethiopia
[3]Center for Public Health and Human Rights, Johns Hopkins Bloomberg School of Public Health, Baltimore, Maryland, USA
[4]Johns Hopkins School of Nursing, Baltimore, Maryland, USA
[5]Center for Global Health, Johns Hopkins University, Baltimore, Maryland, USA
[6]Ethiopia Federal Ministry of Health, Addis Ababa, Ethiopia

**Contributors** SNW designed the study, oversaw study details, drafted the manuscript, and acts as guarantor. RY oversaw all aspects of qualitative training and data collection. AW conducted training and data collection. JQ and RMM conducted qualitative coding. MRD and NG served as ethical supervisors in the project. FS and LZ provided technical insight into Ethiopian IPV response systems. LAZ served as the principal investigator for the parent study and provided technical assistance. All authors participated in writing and approving the final manuscript.

**Funding** Bill& Melinda Gates Foundation (INV 009466), the Johns Hopkins Specialised Centre for Research Excellence in Sex Differences (U54AG062333), and the Foundation for Gender-Specific Medicine. Under the grant conditions of the Bill & Melinda Gates Foundation, a Creative Commons Attribution 4.0 Generic License has already been assigned to the Author Accepted Manuscript version that might arise from this submission.

**Competing interests** None declared.

**Patient and public involvement** Patients and/or the public were involved in the design, or conduct, or reporting, or dissemination plans of this research. Refer to the Methods section for further details.

**Patient consent for publication** Not applicable.

**Ethics approval** Institutional Review Board approval was obtained at both JHSPH (IRB00013278) and AAU College of Health Sciences (077/20/SPH), and protocols were implemented in line with best practices for violence research.

**Provenance and peer review** Not commissioned; externally peer reviewed.

**Data availability statement** Quantitative data are available on request from pmadata.org. Qualitative data are not available to maximise participant confidentiality.

**ORCID iDs**
Shannon N. Wood http://orcid.org/0000-0003-4389-3526
Rachel Mait Milkovich http://orcid.org/0000-0002-0388-0336
Linnea A. Zimmerman http://orcid.org/0000-0002-0118-0889

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
