## [Reviewer comments · BMJ Open]

ARTICLE DETAILS

TITLE (PROVISIONAL)	Impact of the COVID-19 pandemic on intimate partner violence during pregnancy: evidence from a multi-methods study of recently pregnant women in Ethiopia
AUTHORS	Wood, Shannon; Yirgu, Robel; Wondimagegnehu, Abigiya; Qian, Jiage; Milkovich, Rachel; Decker, Michele; Glass, Nancy; Seid, Fatuma; Zekarias, Lensa; Zimmerman, Linnea

VERSION 1 – REVIEW

REVIEWER	Gartland, Deirdre Murdoch Childrens Research Institute, Healthy Mothers Healthy Families
REVIEW RETURNED	25-Oct-2021

GENERAL COMMENTS	Thank you for the opportunity to review this paper. It is very well written and thoughtful paper exploring women's experiences of IPV during pregnancy across Ethiopia, before and during the COVID-19 pandemic. The paper was well written, with a comprehensive and clear background and discussion, and statistical approaches and results very clearly presented and interpreted. I did find the wording "during COVID" confusing. I was unsure to begin with if the women had contracted COVID-19. As COVID-19 is the virus, it might be clearer if it is referred to as "the COVID-19 pandemic"? Also being more specific throughout. e.g "Demographic characteristics of study participants by pre/post exposure are presented in Table 1....." I assume this is exposure to the government-imposed restrictions rather than the virus? It would also be good to provide a bit more detail of the Ethiopian pandemic context for the reader – What were the lockdown restrictions imposed? How long did they last and did this differ region by region? Was there widespread transmission/impacts? What was the government/community response/intervention/support for areas impacted? Study Design I was not clear on whether it was a mixed methods study or a multi-method study (leaning towards the latter). Were the qualitative interviews designed as part of the original study method, or developed during the 'natural experiment' opportunity that arose? Were they directly informed by the quantitative data (apart from targeting women who reported IPV)? Did the qualitative data inform quantitative analyses? Did the interviews explore the women's perception of whether violence had increased during the pandemic restrictions? A bit more detail either way might be helpful. Statistical Analyses
---

	There appears to be a sentence cut off in the second last sentence on page 5, line39. Discussion In high income countries, sexual violence almost always occurs in the context of physical and/or emotional violence. A bit more discussion of these findings could be interesting for readers who are not across this literature – is it a common finding in low/middle income countries?
--	---

REVIEWER	Jack, Susan McMaster University
REVIEW RETURNED	02-Nov-2021

GENERAL COMMENTS	It was a pleasure to review this interesting and timely paper, especially given the availability of a dataset that captures the experiences of almost the full population of pregnant individuals  1. Abstract: please name mixed methods design. What is the purpose/objective for using MM (e.g. to explain, to converge findings?) 2. Add the word "and" between the two study objectives e.g. "quantitative data; and 2) contextualize..." 3. Design - please identify both overarching quantitative (is this a cohort study using a large database) and qualitative study components (e.g, fundamental qualitative description). 4. Abstract line 10-12 - use of word interview for both study designs might lead to confusion; perhaps structured survey administered to women at baseline (** and is this any time during pregnancy???) , 6-weeks, 6-months etc.. . THEN use the term "in-depth semi-structured" interviews for the qual component. 5. One of the reasons why I asked for MM method design to be articulated is I am trying to understand the "function" of the two data sets... was in a convergent design where the postpartum survey and interview were conducted "about the same time" to see how the data compliments, converges, diverges OR did you purposefully do the interviews AFTER the survey b/c you wanted to first analyze the quant data - and then use the qual interviews to "explain" the quant findings?? 6. In abstract you use both COVID and COVID-19, I would recommend selecting the latter and employing this term consistently throughout 7. Article summary - typically the term 'generalizable' is used in reference to quantitative findings; perhaps a more suitable term for your qual findings would be "transferability" 8. An incredibly strong, clear, succinct and engaging introduction. Outstanding. 9. Introduction: small edits - change to "abusive partners may limit access.." and Linkage to Linkages 10. Introduction - purpose statement - can you further clarify the statement "through exploration of lockdown measures" e.g. "through exploration of the impact that local lockdown measures had on X?" And was it only lockdown measures you were looking at impact of or all public health measures aimed to limit social and physical distancing? 11. Typically a mixed methods study will also include language that denotes the MM function of the study - or explains how the two components will be integrated. 12. Methods - study design. I would recommend starting this section by identifying that MM is the overarching design (and state design type - eg. sequential explanatory or convergent design);
--

	then clearly articulate that there are two study components and are they done at the same time or sequentially and why. It then makes sense to transition to discussing the first study component (e.g. the cohort study). If this is a sequential MM study - then it can help to identify study components as Phase 1 and Phase 2. 13. Can you provide additional detail as to when the baseline survey is completed? e.g. anytime during pregnancy, before or at a certain weeks' gestation? 14. With respect to organization of the methods - would it be possible to have a section titled Phase 1 Quantitative - then with subsections for sampling, measures, data collection and statistical analysis. THEN because this is MM - describe if there was an interim/linking step (e.g. were quant data analyzed before the conduct of the qual interviews so that the findings could be explored within the qualitative interviews). THEN have a second section e.g Phase 2 Qualitative. 15. Then in the second qualitative phase - please provide sub-sections for : design, purposeful sampling, data collection and data analysis. 16. Need to identify what design guided qualitative component. From a review of your objective, analysis, findings etc - it seems to me that you used fundamental qualitative description (Bradshaw et al 2017 in the journal Global Qualitative Nursing Research have an excellent overview of this design if you need a reference). 17. For the qualitative component, what was the total "n" of all participants who completed the survey in the two regions? Then of that "n" - how many individuals in each region met the inclusion criteria? Then that will help us to make a bit more sense of the n=14 and n=10. But for the 14 and 10 participants, how did you sample them? Was it random purposeful? Convenience (e.g. they met the inclusion criteria and if they were next on the list and said yes to interview , they were in?). A priori - what was your estimate of the needed sample size (was it 24?). 18. What language were the interviews conducted in? If not in English - at what point did translation/back translation occur? 19. I don't understand what is meant by "coding was complete when saturation of themes was achieved" -- Does this mean you didn't code all data? Typically - my coding is complete when I have open coded every line/section of the transcripts. Also (if one believes in the concept of saturation) - typically it is an indicator that once a researcher determines that the "data are saturated" - one stops recruitment and data collection (not coding). 20. Excellent! Excited to see matrices being used in the analysis - I hope that in the analysis -there will then be some nice compare/contrast across the different types of IPV. 21. A key characteristic of mixed methods research is to have a plan to describe how /when the data will be integrated. Mixed methods is not just about conducting qual and quant components in the same study. So one thing you might consider is a table - with both data types - e.g. IPV data types across the top (quant data) and then down the left column - your broad themes -- and then in the cells a summary of how "experiences" (based on the qual data) differed among different IPV types. THIS is a major recommendation. 22. Page 6 line 46 ... Is it that COVID-19 affected IPV -- or is it that the "lockdown measures" implemented to mitigate the spread/transmission of COVID-19 impacted ...?? 23. For the presentation of your qualitative findings, it might be helpful to have a framework or outline of key findings - or even in the text some sub-headings (from your categories), I am not sure
--	--

	your findings have reached the levels of themes -- but that is okay for descriptive qual research). Then organize the findings by your sub-headings. 24. One strategy to increase the depth of your analysis - and to increase degree of interpretation (and to allow the reader to do less work) - is to NOT just string quotes together (eg..page 7). But instead for each quote, highlight what finding is uniquely illustrated by the quote - then contextualize the quote by a brief introduction e.g. As one woman, who lived in a rural community with 3 children, explained... 25. In your qualitative data analysis matrix (see above), I would also be interested to see the compare /contrast of experiences by IPV type and geography (rural/urban) since you discuss it in your discussion. 26. Limitations. In your first limitation- this makes me go back and ask the question "Were you trying to measure pre/post exposure to COVID-19 (e.g., the actual virus/illness) or is it exposure to the public health social/physical measures (e.g. lockdown) implemented to mitigate the spread of the virus ?" (it seems it is the latter to me -- so perhaps some increased attention to language throughout the manuscript is warranted). 27. Limitations - if you provide a rich description of the two regions and a good rationale for why qual data were only conducted in those two regions then that helps the reader with transferability of findings. Generalizability is a term associated with quant paradigm. So perhaps speak to transferability of findings ...e.g. these qualitative findings may only be transferable to other contexts where (....)... 28. Reference list needs substantive editing to ensure that all references are cited consistently and as per required writing style
--	---

REVIEWER	Teshome , Abel St Paul's Hospital Millennium Medical College, Department of Obstetrics and Gynecology
REVIEW RETURNED	04-Nov-2021

GENERAL COMMENTS	Reviewer comments Result section  1. Please report prevalence in n(%) format consistently throughout your result section. You are only reporting the frequency(%) 2. Please write the odds ratio with the comparator(Paper page(pp)6, Lines 33-40) , e.g. 'pregnant women have a 50% increment in IPV at time of covid 19 compared to pre-covid time(aOR=1.51;95% C.I=1.06-2.15; P=0.02) " 3. On statistical analysis better to have a Bivariate logistic regression table with Crude Odds ratios and P-value. Then you can decide to take those predictors with p-value less than 0.05 to be included in the multivariate logistic regression and report the adjusted odds ratio with its P-value on the additional table(Table 3) 4. There is no need to report on statistically non-significant with its adjusted odds ratio unless it is unexpected finding e.g. the report on adjusted odds ratio for rural women(pp 6, Lines 38-39) 5. You need to avoid vague terms like consistently high. You need to report on the exact number with prevalence 6. Again please avoid vague terms like "significant overall increase" , as I said it please report the magnitude of increase when you report the adjusted odds ratio. For example, " there is a 60% increase in odds of physical violence during covid compared to pre-covid time((aOR=2.46; 95% CI=1.18-5.10; p=0.02)" Discussion
---

	7. The first paragraph of your discussion section should be a summary of your key findings, which you will compare and contrasts them in the subsequent paragraphs of your discussion 8. There are recommendations in your discussion which is wrong. In your discussion section what you need to do is compare and contrast your findings with previous research findings and try to explain if your finding is different from the previous research findings. For example, this statement “Continuous monitoring within national surveillance systems and large-scale cohort studies can assist identifying who is at risk for IPV during most recent pregnancy” 9. Recommendations and conclusion should be written at the last paragraph/s of the discussion 10. Better to compare and contrast predictors which are found to be significant when you do adjusted odds ratio. Non-significant predictors can also be included if the finding is unexpected
--	--

VERSION 1 – AUTHOR RESPONSE

Authors’ Response: Thank you for your positive review of our manuscript. We are pleased to provide a revision that addresses the thoughtful reviewer feedback. Please see specific points below.

Reviewer: 1: Dr. Deirdre Gartland, Murdoch Childrens Research Institute

Comments to the Author: Thank you for the opportunity to review this paper. It is very well written and thoughtful paper exploring women’s experiences of IPV during pregnancy across Ethiopia, before and during the COVID-19 pandemic. The paper was well written, with a comprehensive and clear background and discussion, and statistical approaches and results very clearly presented and interpreted.

Authors’ Response: Many thanks for your constructive review of our manuscript.

- I did find the wording “during COVID” confusing. I was unsure to begin with if the women had contracted COVID-19. As COVID-19 is the virus, it might be clearer if it is referred to as ‘the COVID-19 pandemic’? Also being more specific throughout. e.g “Demographic characteristics of study participants by pre/post exposure are presented in Table 1.....” I assume this is exposure to the government-imposed restrictions rather than the virus?

Authors’ Response: Thank you for these suggestions to clarify timepoints. We have clarified “during the COVID-19 pandemic” when possible. We further clarify that the pre/post exposure is in relation to government-imposed restrictions in Ethiopia, as outlined on pg. 5, lines 26-28.
- It would also be good to provide a bit more detail of the Ethiopian pandemic context for the reader – What were the lockdown restrictions imposed? How long did they last and did this differ region by region? Was there widespread transmission/impacts? What was the government/community response/intervention/support for areas impacted?

Authors’ Response: We agree it is important to contextualize the pandemic within Ethiopia and have included these details accordingly (pg. 4, lines 18-22). We clarify that lockdown restrictions included, but were not limited to, instituting physical distancing, school closures, prohibition of in-house visitation outside of family members, capacity limits and masking requirements on public transportation, and discouragement of within-country travel. While these restrictions were imposed country-wide, urban residents may have been

disproportionately impacted due to population density, transportation infrastructure, and health-seeking behavior patterns.

3. Study Design--I was not clear on whether it was a mixed methods study or a multi-method study (leaning towards the latter). Were the qualitative interviews designed as part of the original study method, or developed during the 'natural experiment' opportunity that arose? Were they directly informed by the quantitative data (apart from targeting women who reported IPV)? Did the qualitative data inform quantitative analyses? Did the interviews explore the women's perception of whether violence had increased during the pandemic restrictions? A bit more detail either way might be helpful.

Authors' Response: We agree with these points and have accordingly reframed as a multi-methods study, as well as provided additional details on the intersection between quantitative and qualitative data (pg. 4, lines 49-50). We clarify that the qualitative interviews were not designed as part of the parent study, and instead developed during the natural experiment and designed to be complementary—these interviews were not informed by the quantitative data, and vice versa, thus constituting as a multi-methods study. The interviews explored women's personal experiences with violence before and/or during the pandemic restrictions.

4. Statistical Analyses--There appears to be a sentence cut off in the second last sentence on page 5, line 39.

Authors' Response: Thank you—this sentence has been revised (pg. 5, line 40).

5. Discussion--In high income countries, sexual violence almost always occurs in the context of physical and/or emotional violence. A bit more discussion of these findings could be interesting for readers who are not across this literature – is it a common finding in low/middle income countries?

Authors' Response: We agree that this is an important point and have added additional details within the Discussion (pg. 8, lines 24-27). We further clarify that this is not a common finding, and that Ethiopia is one of the few settings where the prevalence of sexual violence is higher than that of physical violence.

Reviewer: 2: Dr. Susan Jack, McMaster University

Comments to the Author: It was a pleasure to review this interesting and timely paper, especially given the availability of a dataset that captures the experiences of almost the full population of pregnant individuals.

Authors' Response: Many thanks for your review of our manuscript and valuable feedback.

Abstract

6. Please name mixed methods design. What is the purpose/objective for using MM (e.g. to explain, to converge findings?)

Authors' Response: Thank you for this question—per Reviewer 1's comment, we agree that this is a multi-methods design and clarify that the purpose of the qualitative was to contextualize our results given relatively limited quantitative measures. These phases were not designed together or integrated—we have added details to clarify our study design (pg. 4, lines 49-50).

7. Add the word "and" between the two study objectives e.g. "quantitative data; and 2) contextualize..."

Authors' Response: We have added "and" between the two study objectives within the Abstract (pg. 2, line 3).

8. Design - please identify both overarching quantitative (is this a cohort study using a large database) and qualitative study components (e.g, fundamental qualitative description).
Authors' Response: We have added these components to the Abstract (pg. 2, line 7).
9. Abstract line 10-12 - use of word interview for both study designs might lead to confusion; perhaps structured survey administered to women at baseline (** and is this any time during pregnancy???), 6-weeks, 6-months etc.. . THEN use the term "in-depth semi-structured" interviews for the qual component.
Authors' Response: We agree that this is confusing and have accordingly used "survey" to identify the quantitative portion, whereas "interview" identifies the qualitative portion (pg. 2, line 9). We clarify that the baseline interview could have occurred at any point during pregnancy or within 6-weeks postpartum. These procedures are further outlined in the protocol paper for the parent study (Zimmerman et al.).
10. One of the reasons why I asked for MM method design to be articulated is I am trying to understand the "function" of the two data sets... was in a convergent design where the postpartum survey and interview were conducted "about the same time" to see how the data compliments, converges, diverges OR did you purposefully do the interviews AFTER the survey b/c you wanted to first analyze the quant data - and then use the qual interviews to "explain" the quant findings??
Authors' Response: We clarify that the qualitative portion was an additional component that served as explanatory, however, given that the study was not initially designed with both data sources in mind, we believe that it is more appropriate to identify our design as multi-methods and have updated this language throughout.
11. In abstract you use both COVID and COVID-19, I would recommend selecting the latter and employing this term consistently throughout.
Authors' Response: Per this reviewer's suggestion, we have utilized "COVID-19" throughout.

Article Summary

12. Typically, the term 'generalizable' is used in reference to quantitative findings; perhaps a more suitable term for your qual findings would be "transferability"
Authors' Response: We agree with this reviewer's point and have edited "generalizable" to "transferable" (pg. 3, line 11).

Introduction

13. An incredibly strong, clear, succinct and engaging introduction. Outstanding.
Authors' Response: Thank you so much—your kindness is greatly appreciated.
14. Small edits - change to "abusive partners may limit access.." and Linkage to Linkages
Authors' Response: Thank you for these suggestions—we have updated accordingly (pg. 4, line 10; pg. 4, line 13).
15. Purpose statement - can you further clarify the statement "through exploration of lockdown measures" e.g. "through exploration of the impact that local lockdown measures had on X?" And was it only lockdown measures you were looking at impact of or all public health measures aimed to limit social and physical distancing?

Authors' Response: Thank you for this question. As the in-depth interviews probed more broadly on women's experience of IPV during COVID than just lockdown or public health measures, we have chosen to omit specificity to lockdown measures (pg. 4, line 37).

16. Typically a mixed methods study will also include language that denotes the MM function of the study - or explains how the two components will be integrated.

Authors' Response: We have clarified these details of our multi-methods study (pg. 4, lines 49-50).

Methods

17. Study design. I would recommend starting this section by identifying that MM is the overarching design (and state design type - eg. sequential explanatory or convergent design); then clearly articulate that there are two study components and are they done at the same time or sequentially and why. It then makes sense to transition to discussing the first study component (e.g. the cohort study). If this is a sequential MM study - then it can help to identify study components as Phase 1 and Phase 2.

Authors' Response: Thank you for this suggestion. We have reorganized to clarify the overall study design first (multi-methods) and then gone on to explain its components.

18. Can you provide additional detail as to when the baseline survey is completed? e.g. anytime during pregnancy, before or at a certain weeks' gestation?

Authors' Response: We clarify that the baseline survey could be completed any time during pregnancy (pg. 4, line 44). To overcome limitations surrounding differences in gestational age and exposure to IPV during pregnancy, we utilized IPV data from the six-weeks postpartum interview, where the survey items are specific to violent behaviors at *any point during pregnancy*.

19. With respect to organization of the methods - would it be possible to have a section titled Phase 1 Quantitative - then with subsections for sampling, measures, data collection and statistical analysis. THEN because this is MM - describe if there was an interim/linking step (e.g. were quant data analyzed before the conduct of the qual interviews so that the findings could be explored within the qualitative interviews). THEN have a second section e.g Phase 2 Qualitative.

Authors' Response: Thank you for these organization suggestions. We have updated the section headings to describe the Quantitative or Qualitative phases more thoroughly. Given the multi-methods nature of the design, we omitted the interim/linking step.

20. Then in the second qualitative phase - please provide sub-sections for : design, purposeful sampling, data collection and data analysis.

Authors' Response: We have also updated the qualitative phase headings.

21. Need to identify what design guided qualitative component. From a review of your objective, analysis, findings etc - it seems to me that you used fundamental qualitative description (Bradshaw et al 2017 in the journal Global Qualitative Nursing Research have an excellent overview of this design if you need a reference).

Authors' Response: Thank you for this resource and we agree the provided reference is excellent—we have accordingly incorporated the qualitative descriptive design approach and reference (pg. 5, line 45).

22. For the qualitative component, what was the total "n" of all participants who completed the survey in the two regions? Then of that "n" - how many individuals in each region met the inclusion criteria? Then that will help us to make a bit more sense of the n=14 and n=10. But

for the 14 and 10 participants, how did you sample them? Was it random purposeful? Convenience (e.g. they met the inclusion criteria and if they were next on the list and said yes to interview, they were in?). A priori - what was your estimate of the needed sample size (was it 24?).

Authors' Response: We clarify that we utilized random purposive sampling. If participants met the inclusion criteria, we randomly selected them from the sampling frame. This sampling frame included 17 IPV survivors in Oromiya and 18 IPV survivors in SNNP; we have included these components within the manuscript (pg. 6, lines 4-6). We further clarify that our a priori sample size was 30, however, as themes repeated during interviews, we concluded at 24 due to feasibility considerations and impending violence within the study regions.

23. What language were the interviews conducted in? If not in English - at what point did translation/back translation occur?

Authors' Response: Thank you for this question. Interviews were conducted in Amharic, Afan Oromo, or local languages within SNNP region. We clarify that over 45 languages are spoken with SNNP, the most common being Sidama, Guragigna, and Wolayta. As such, local translators were used for both the quantitative and qualitative portions within this region. These details are included for the quantitative portion within the cited protocol for PMA Ethiopia and we have added these details to the Methods section for the qualitative portion (pg. 6, lines 14-15).

24. I don't understand what is meant by "coding was complete when saturation of themes was achieved" -- Does this mean you didn't code all data? Typically - my coding is complete when I have open coded every line/section of the transcripts. Also (if one believes in the concept of saturation) - typically it is an indicator that once a researcher determines that the "data are saturated" - one stops recruitment and data collection (not coding).

Authors' Response: We clarify that we used inductive thematic analysis and coded all data. This wording was intended to convey that we did not add any additional codes to the codebook once we felt that the themes were saturated.

25. Excellent! Excited to see matrices being used in the analysis - I hope that in the analysis - there will then be some nice compare/contrast across the different types of IPV.

Authors' Response: Thank you—we are also excited to utilize a matrix approach.

26. A key characteristic of mixed methods research is to have a plan to describe how /when the data will be integrated. Mixed methods is not just about conducting qual and quant components in the same study. So one thing you might consider is a table - with both data types - e.g. IPV data types across the top (quant data) and then down the left column - your broad themes -- and then in the cells a summary of how "experiences" (based on the qual data) differed among different IPV types. THIS is a major recommendation.

Authors' Response: Thank you for this recommendation. We agree that it's useful to integrate quantitative and qualitative components when using mixed-methods research. We have reframed the paper to describe as multi-methods, as the quantitative and qualitative methodology and design was not fully integrated. Given the limited quantitative measures to explore IPV during COVID in comparison to the extensive qualitative descriptions and this reframing as multi-methods, we would like to retain our initial structure, however, ultimately defer to the Editorial team.

27. Page 6 line 46 ... Is it that COVID-19 affected IPV -- or is it that the "lockdown measures" implemented to mitigate the spread/transmission of COVID-19 impacted ...??

Authors' Response: We have updated this line to read "the COVID-19 pandemic and implementation of local lockdown measures" (pg. 7, line 10).

28. For the presentation of your qualitative findings, it might be helpful to have a framework or outline of key findings - or even in the text some sub-headings (from your categories), I am

not sure your findings have reached the levels of themes -- but that is okay for descriptive qual research). Then organize the findings by your sub-headings.

Authors' Response: Thank you for this suggestion. Given the multi-methods nature of this research, we have opted to not include a framework within the present paper. We clarify that these sub-themes are useful to contextualize the quantitative data within the present manuscript and that additional manuscripts have further explored other aspects of our qualitative data (i.e. help-seeking behaviors).

29. One strategy to increase the depth of your analysis - and to increase degree of interpretation (and to allow the reader to do less work) - is to NOT just string quotes together (eg..page 7). But instead for each quote, highlight what finding is uniquely illustrated by the quote - then contextualize the quote by a brief introduction e.g. As one woman, who lived in a rural community with 3 children, explained...

Authors' Response: Thank you for this suggestion—we agree that these details are useful to contextualize and have included throughout the Results.

30. In your qualitative data analysis matrix (see above), I would also be interested to see the compare /contrast of experiences by IPV type and geography (rural/urban) since you discuss it in your discussion.

Authors' Response: We agree that it would be help to compare/contrast experience of IPV type and geography within the qualitative section, however, unfortunately do not have sufficient granularity in experiences to do so. As purposive sampling did not necessitate urban and rural geography, the majority of participants within these regions are from agrarian communities. Further, many women describe experiencing multiple types of IPV. We feel that the quantitative data are more useful to disaggregating these experiences.

31. Limitations. In your first limitation- this makes me go back and ask the question "Were you trying to measure pre/post exposure to COVID-19 (e.g., the actual virus/illness) or is it exposure to the public health social/physical measures (e.g. lockdown) implemented to mitigate the spread of the virus ?" (it seems it is the latter to me -- so perhaps some increased attention to language throughout the manuscript is warranted).

Authors' Response: We clarify that we were attempting to measure exposure to the public health/lockdown measures implemented to mitigate the spread of the virus. We have adjusted the language throughout to increase clarity.

32. Limitations - if you provide a rich description of the two regions and a good rationale for why qual data were only conducted in those two regions then that helps the reader with transferability of findings. Generalizability is a term associated with quant paradigm. So perhaps speak to transferability of findings ...e.g. these qualitative findings may only be transferable to other contexts where (....)...

Authors' Response: Thank you for this suggestion. We clarify that qualitative data were collected in these two regions due to high IPV during pregnancy and feasibility (pg. 5, line 48). We have further updated this language to speak to transferability of findings, rather than generalizability (pg. 9, line 4).

33. Reference list needs substantive editing to ensure that all references are cited consistently and as per required writing style

Authors' Response: Thank you—we have edited the reference list accordingly.

Reviewer: 3: Dr. Abel Teshome, St Paul's Hospital Millennium Medical College

Authors' Response: Thank you for your thorough review of our manuscript and helpful edits.

Result section

34. Please report prevalence in n (%) format consistently throughout your result section. You are only reporting the frequency (%)
Authors' Response: Thank you for this suggestion. To maximize clarity for multiple measures and timepoints, we only present frequency within the prose of the results, however, the n's are available within the Tables. We defer to the Editorial team in whether they would like n's included within the prose.
35. Please write the odds ratio with the comparator (Paper page(pp)6, Lines 33-40), e.g. 'pregnant women have a 50% increment in IPV at time of covid 19 compared to pre-covid time(aOR=1.51;95% C.I=1.06-2.15; P=0.02) ”
Authors' Response: We have cross-checked to ensure that the referent group is included with odds ratio presentation (pg. 6, lines 44-pg. 7, line 7).
36. On statistical analysis better to have a Bivariate logistic regression table with Crude Odds ratios and P-value. Then you can decide to take those predictors with p-value less than 0.05 to be included in the multivariate logistic regression and report the adjusted odds ratio with its P-value on the additional table (Table 3)
Authors' Response: We clarify that the bivariate logistic regression table is not presented as there were minimal differences between the unadjusted and adjusted odds ratios. We have added this detail to the Methods to ensure that the reader understands the process and that the bivariate analysis was run prior to adjustment, although it is not presented (pg. 5, line 37).
37. There is no need to report on statistically non-significant with its adjusted odds ratio unless it is unexpected finding e.g., the report on adjusted odds ratio for rural women (pp 6, Lines 38-39)
Authors' Response: We agree that we would generally not report statistically non-significant results, however, we do feel that these results are unexpected and have opted to retain within the prose.
38. You need to avoid vague terms like consistently high. You need to report on the exact number with prevalence
Authors' Response: We have increased specificity when reporting prevalence (pg. 7, lines 1-2).
39. Again please avoid vague terms like “significant overall increase”, as I said it please report the magnitude of increase when you report the adjusted odds ratio. For example, “ there is a 60% increase in odds of physical violence during covid compared to pre-covid time((aOR=2.46; 95% CI=1.18-5.10; p=0.02)”
Authors' Response: We have increased specificity to report magnitude of increase (pg. 7, lines 4-5).

Discussion

40. The first paragraph of your discussion section should be a summary of your key findings, which you will compare and contrasts them in the subsequent paragraphs of your discussion.
Authors' Response: Thank you—we confirm that we have included a summary of our key findings within the first paragraph of the Discussion.
41. There are recommendations in your discussion which is wrong. In your discussion section what you need to do is compare and contrast your findings with previous research findings and try to explain if your finding is different from the previous research findings. For example,

this statement “Continuous monitoring within national surveillance systems and large-scale cohort studies can assist identifying who is at risk for IPV during most recent pregnancy”

Authors’ Response: Thank you for this comment, however, we respectfully disagree and seek to retain recommendations within our Discussion. While we agree with this reviewer that comparing and contrasting findings to other research is a best practice, such comparisons are challenging to make in this instance due to the dearth of population-based data specific to IPV during COVID-19 in Ethiopia. Additionally, we refer to the submission guidelines of BMJ Open, which state that the conclusions should focus on the “primary conclusions and their implications, suggest areas for further research in appropriate”(available at https://bmjopen.bmj.com/pages/authors/#submission_guidelines). We feel that these results are helpful for both researchers and violence prevention and response programs and highlight their implications for these stakeholders. We cautiously provide recommendations based on the current data, and urge continued monitoring as the COVID-19 pandemic continues to unfold.

42. Recommendations and conclusion should be written at the last paragraph/s of the discussion
Authors’ Response: Thank you for this suggestion. We clarify that our recommendations for IPV prevention and response efforts are largely focused within the last paragraph beginning with “We offer two concrete recommendations for maternal health providers” (pg. 9, line 9).

43. Better to compare and contrast predictors which are found to be significant when you do adjusted odds ratio. Non-significant predictors can also be included if the finding is unexpected
Authors’ Response: Many thanks for this comment, however, we are unclear what this reviewer is asking.

VERSION 2 – REVIEW

REVIEWER	Jack, Susan McMaster University
REVIEW RETURNED	03-Feb-2022

GENERAL COMMENTS	I enjoyed the opportunity to continue to review this paper - and appreciate the use of multiple methods; creates a much more comprehensive picture of women's experiences of violence during pregnancy. Two small minor revisions 1) - page 5, line 48 -- it is not interviews that are purposefully sampled -- people (not interviews) are recruited (sampled) into a study. If you could just revise this sentence that would be great. E..g. A purposeful sample of 24 women identified through the survey data to have experienced IPV were invited to participate in this qualitative components. (then continue on with rest of paragraph)... 2) Page 6, line 22 - please omit term emerge. Findings do not emerge from data, they are constructed, developed etc (please see extensive writings of Braun & Clarke - leaders in thematic qualitative analysis - who have lengthy discussions that "themes do not emerge")
---

VERSION 2 – AUTHOR RESPONSE

Authors' Response: Thank you for your continued positive review of our manuscript. We are pleased to provide a revision that addresses additional reviewer feedback. Please see specific points below.

Reviewer 2: Dr. Susan Jack, McMaster University

Comments to the Author: I enjoyed the opportunity to continue to review this paper - and appreciate the use of multiple methods; creates a much more comprehensive picture of women's experiences of violence during pregnancy.

Authors' Response: Many thanks for your continued review and suggestions to improve our manuscript.

Two small minor revisions:

1. page 5, line 48 -- it is not interviews that are purposefully sampled -- people (not interviews) are recruited (sampled) into a study. If you could just revise this sentence that would be great. E..g. A purposeful sample of 24 women identified through the survey data to have experienced IPV were invited to participate in this qualitative components. (then continue on with rest of paragraph)...

Authors' Response: We have revised this sentence in line with this reviewer's suggested wording (pg. 5, line 46).

2. Page 6, line 22 - please omit term emerge. Findings do not emerge from data, they are constructed, developed etc (please see extensive writings of Braun & Clarke - leaders in thematic qualitative analysis - who have lengthy discussions that "themes do not emerge").

Authors' Response: Thank you for this suggestion and citation—we have deleted “emergent” accordingly (pg. 6, line 20).